# Trends in Self-Rated Oral Health and Its Associations with Oral Health Status and Oral Health Behaviors in Japanese University Students: A Cross-Sectional Study from 2011 to 2019

**DOI:** 10.3390/ijerph192013580

**Published:** 2022-10-20

**Authors:** Momoko Nakahara, Naoki Toyama, Daisuke Ekuni, Noriko Takeuchi, Takayuki Maruyama, Aya Yokoi, Daiki Fukuhara, Nanami Sawada, Yukiho Nakashima, Manabu Morita

**Affiliations:** 1Department of Preventive Dentistry, Okayama University Graduate School of Medicine, Dentistry and Pharmaceutical Sciences, Okayama 700-8558, Japan; 2Department of Preventive Dentistry, Academic Field of Medicine, Dentistry and Pharmaceutical Sciences, Okayama University, Okayama 700-8558, Japan; 3Department of Preventive Dentistry, Okayama University Hospital, Okayama 700-8558, Japan

**Keywords:** self-rated oral health, oral health behaviors, caries, gingivitis, oral hygiene, oral health, behavioral sciences

## Abstract

Self-rated oral health (SROH) is a valid, comprehensive indicator of oral health status. The purpose of this cross-sectional study was to analyze how oral health behaviors and clinical oral status were associated with SROH and how they had changed over the course of nine years in Japanese university students. Data were obtained from 17,996 students who underwent oral examinations and completed self-questionnaires from 2011 to 2019. Oral status was assessed using the decayed and filled teeth scores, bleeding on probing (BOP), probing pocket depth, the Oral Hygiene Index-Simplified (OHI-S), oral health behaviors, and related factors. SROH improved from 2011 to 2019. The logistic regression model showed that university students who were female and had a high daily frequency of tooth brushing, no BOP, no decayed teeth, no filled teeth, and a low OHI-S score and were significantly more likely to report very good, good, or fair SROH. An interaction effect was observed between survey year and regular dental check-ups (year × regular dental check-ups). The improvement trend in SROH might be associated with changes in oral health behaviors and oral health status.

## 1. Introduction

Self-rated oral health (SROH) is a valid, comprehensive indicator of overall oral health status, and it is often assessed in epidemiological studies [1,2,3]. In recent years, it has been recognized that clinical measures of oral health should be supplemented by subjective assessment of oral symptoms to improve both the quality of dental care and people’s satisfaction [4,5]. SROH is a relatively simple indicator and may be an easy and useful method for collecting dental information and planning health promotion interventions [1,2,3,6,7,8]. Based on the results of previous studies, surveillance of oral health should include SROH [1,9,10].

Previous studies reported that SROH was associated with a variety of factors, including demographic, socioeconomic, and behavioral factors, objective oral health, and oral health awareness [11]. SROH was also associated with clinical conditions [decayed, missing, and filled teeth (DMFT), gingival bleeding, and periodontal pockets], and oral symptoms (oral pain, tooth sensitivity, bad breath, and difficulty in speaking) [1,12,13,14,15,16,17,18]. Several oral health behaviors such as dental attendance and tooth brushing were associated with SROH [12,18,19]. Furthermore, poor SROH was a predictor of five-year and ten-year tooth loss [20].

In recent years, dental caries and oral health behaviors have been getting better in the Japanese population based on the National Survey of Dental Diseases in Japan [21,22]. The proportion of people aged 15–19 years with decayed teeth and the average DMFT score of people aged 15–24 years have decreased from 2011 to 2016, especially in young adults [21,22]. In addition, the percentage of people aged 1 to 85 and over who brushed their teeth more than once a day has increased from 2011 to 2016. Thus, SROH might be getting better over time. However, few studies have investigated recent changes and trends in SROH in Japanese young adults.

Many young university students live away from parental supervision for the first time. Therefore, they can easily change their health and lifestyle behaviors [1,9]. University students’ household type was directly associated with regular dental check-ups and indirectly contributed to gingival status [23]. Some previous studies reported that oral health behaviors such as tooth brushing, use of dental floss, and regular dental check-ups affected oral hygiene and periodontal status in university students and adults [24,25]. Poor oral health behaviors related to tooth brushing and dental check-ups can easily cause poor clinical status, which may lead to a vicious circle that negatively affects SROH.

We hypothesized that SROH changes over time, and oral health behaviors and oral health status influence changes in SROH of Japanese university students. The purpose of this cross-sectional study was to analyze the trends in SROH and how oral health behaviors and oral health status are associated with changes in the SROH of Japanese university students.

## 2. Materials and Methods

### 2.1. Ethics Statement

The protocol of this study was approved by the Ethics Committee of Okayama University Graduate School of Medicine, Dentistry, and Pharmaceutical Sciences (No. 1060). All methods were performed in accordance with the Declaration of Helsinki. Informed consent was obtained verbally from each participant. This study conformed with the Strengthening the Reporting of Observational Studies in Epidemiology (STROBE) guidelines.

### 2.2. Study Population

First-year students volunteered to undergo oral examinations and answer self-administered questionnaires at Okayama University in April from 2011 to 2019. Participants were recruited from all faculties (Faculties of Letters, Education, Law, Economics, Science, Pharmaceutical Sciences, Engineering, Environmental Science and Technology, Agriculture, Medicine, and Dentistry). Inclusion criteria were: students who agreed to participate in the study; students who provided complete data for both the oral examination and the questionnaires; and students who were 18 or 19 years old, in order to avoid age-related factors [1,2,9,25]. Exclusion criteria were students aged ≥ 20 years old and those who provided incomplete data. It was widely promoted to students to undergo oral examinations each year, and all participants who matched the inclusion criteria were included to minimize selection bias.

### 2.3. Self-Questionnaires

The self-administered questionnaire was delivered to each student before participating in oral examinations. In addition to sex and age, the self-administered questionnaires included the following items. The students were asked about the following oral health behaviors: daily frequency of tooth brushing (≥two times/≤one time); use of dental floss (yes/no); and undergoing regular dental check-ups (yes/no) [1,2,9,25]. The SROH was assessed by responses to the question: “In general, how do you consider your oral health?”. The participants chose one of the following responses: “very good”, “good”, “fair”, “poor”, or “very poor” [1,2,9]. The “very good”, “good”, and “fair” responses were combined to represent “good SROH”, and the “poor” and “very poor” responses were combined to represent “poor SROH” [2].

### 2.4. Oral Examination

Nine qualified dentists (D.E., N.T. (Noriko Takeuchi), T.M., A.Y., D.F., N.T. (Naoki Toyama), N.S., Y.N., and M.N.) conducted oral examinations. Dental caries status was evaluated using the decayed teeth and filled scores [26]. Periodontal status was assessed using the community periodontal index (CPI) [26]. For periodontal examination, the following ten teeth were selected: two molars in each posterior sextant and the upper right and lower left central incisors. A CPI probe (YDM, Tokyo, Japan) was used to measure each tooth at six sites (mesiobuccal, mid-buccal, distobuccal, distolingual, mid-lingual, and mesiolingual) for measurement of probing pocket depth (PPD) and bleeding on probing (BOP). Periodontal disease was defined as PPD ≥ 4 mm (CPI pocket score = 1 or 2). The level of dental plaque and calculus was assessed using the Oral Hygiene Index-Simplified (OHI-S) [27]. The OHI-S was dichotomized with a median of 0.33 (good ≤ 0.33; poor > 0.33) [28]. Calibration among the nine dentists was performed before the oral examinations, and the intra- and inter-examiner reliabilities as evaluated by κ statistics were both > 0.8.

### 2.5. Statistical Analyses

SPSS (version 25; IBM, Tokyo, Japan) was used for all statistical analyses, with a two-sided *p*-value < 0.05 considered to indicate significance. First, using the combined data from 2011 to 2019, significant differences between the good SROH group and the poor SROH group were determined by a chi-squared test. Second, odds ratios (ORs) and 95% confidence intervals (CIs) were calculated using a logistic regression model (a forward selection method) [29], to evaluate the associations of good SROH with categorical factors: sex, daily frequency of tooth brushing, use of dental floss, regular dental check-ups, BOP, PPD, the numbers of decayed and filled teeth, and the OHI-S. These independent variables were selected based on the *p*-value (<0.20) on the chi-squared test, because it has been suggested that potential confounders should be eliminated only for a *p*-value > 0.20 to prevent residual confounding [30]. Third, associations of good SROH with seven categorical factors were investigated on a yearly basis based on the result of the second analysis. Finally, associations of good SROH with these seven categorical factors were confirmed considering the survey year effect and their possible interactions with survey year (year × sex, year × daily frequency of tooth brushing, year × regular dental check-ups, year × BOP, year × decayed teeth, year × filled teeth, and year × OHI-S). This analysis investigated whether factor effects related to SROH changed over time across the surveys [29]. The survey years from 2011 to 2019 were coded from one to nine and considered continuous variables.

## 3. Results

A total of 20,090 students underwent voluntary oral examinations and answered self-administered questionnaires from 2011 to 2019. Of these students, 2094 were excluded for the following reasons: age ≥ 20 years (*n* = 1003) and providing incomplete data (*n* = 1091). Finally, the data of 17,996 students (89.6%) were analyzed.

Table 1 shows the characteristics of the participants. This study analyzed data from 17,996 participants (10,142 males and 7854 females; age 18–19 years). Approximately 2000 students participated each year. Proportions of “very good”, “good”, “fair”, “poor” and “very poor” for SROH tended to improve from 4.7%, 17.8%, 51.4%, 22.4%, and 3.7%, respectively, in 2011 (*n* = 2077) to 15.1%, 28.5%, 41.7%, 13.1%, and 1.6%, respectively, in 2019 (*n* = 2019) (Figure 1). In addition, an improving trend was seen in the proportions of students who had a high daily frequency of tooth brushing (81.1% in 2011 and 89.8% in 2019), used dental floss (4.9% in 2011 and 28.9% in 2019), and underwent regular dental check-ups (14.4% in 2011 and 33.3% in 2019). The improving trend was also seen in the proportions of students without BOP (23.4% in 2011 and 26.6% in 2019), decayed teeth (83.7% in 2011 and 91.8% in 2019), and filled teeth (44.2% in 2011 and 56.5% in 2019). The OHI-S median score also improved from 0.67 in 2011 to 0.33 in 2019.

Table 2 shows the comparison between the good SROH group and the poor SROH group. The good SROH group included 14,671 (81.5%) participants. Significant differences in sex, daily frequency of tooth brushing, use of dental floss, regular dental check-ups, BOP, decayed teeth, filled teeth, and OHI-S were observed between the two groups (all *p* < 0.01). Students who were female, had a high daily frequency of tooth brushing, used dental floss, underwent regular dental check-ups, had no BOP, no decayed teeth, no filled teeth, and a low OHI-S score reported good SROH significantly more than the others.

As shown in Table 3, the logistic regression analysis showed that students who were female had a high daily frequency of tooth brushing, underwent regular dental check-ups, had no BOP, no decayed teeth, no filled teeth, and a low OHI-S score reported good SROH significantly more than the others (all *p* < 0.01). Table 4 shows that good SROH was significantly associated with daily frequency of tooth brushing, decayed teeth, filled teeth, and OHI-S (all *p* < 0.05), which was consistent in each year. On the other hand, sex, regular dental check-ups, and BOP showed inconsistent results depending on the survey year.

Table 5 shows that students who were female, had a high daily frequency of tooth brushing, had no BOP, no decayed teeth, no filled teeth, and a low OHI-S score reported good SROH significantly more than the others (all *p* < 0.01). In addition, good SROH was significantly associated with survey year. The result indicates that the more recent the survey year, the more students reported good SROH.

To investigate whether the factors related to SROH changed over time, interaction effects between factors and survey year were confirmed. An interaction effect between survey year and regular dental check-ups (year × regular dental check-ups: *p* = 0.002) was observed. Therefore, predicted probabilities were plotted with the interaction effect in Figure 2. The interaction effect between regular dental check-ups and survey year showed that the regular dental check-up effect varied according to the survey year. Students undergoing regular dental check-ups had higher probabilities of reporting good oral health than those not. The difference between students undergoing regular dental check-ups and those not decreased from 2011 to 2019.

## 4. Discussion

In the present study, the percentage of good SROH students increased over the last nine years. In addition, an interaction effect between survey year and regular dental check-ups (year × regular dental check-ups) was observed in university students. Students who were female, had a high daily frequency of tooth brushing, had no BOP, no decayed teeth, no filled teeth, and a low OHI-S score reported good SROH significantly more than other students. To the best of our knowledge, this was the first cross-sectional study to investigate the trends in SROH and how oral health behaviors and oral health status are associated with changes in SROH of Japanese university students. The improvement trend in SROH might be associated with changes in oral health behaviors and oral health status.

In 2019, 1721 (85.2%) participants had very good, good or fair SROH. According to recent studies [11,31], this value was higher than the data of people aged 20–29 years old in the United States in 2017–2018 (75.3%), aged 20–29 years old in Korea in 2016–2018 (78.0%), and aged 15–34 years old in Australia in 2017–2018 (81.6%). This discrepancy might be due to the differences in sample sizes and age groups. In addition, these studies categorized excellent, very good, and good into good SROH and fair and poor into poor SROH in the United States and Australia [11,31]. However, the present study categorized very good, good, and fair into good SROH based on a previous study of Japanese university students [2]. On the other hand, the present results were similar to the overall results of the study in Canada in 2013–2014 (85.2%), although the subgroup (15–19 years old) of Canadian people showed a relatively higher percentage of good SROH (92.2%) [32]. Taken together, the percentage of good SROH people in the present study was within the same range of the other developed countries.

An improving trend was also seen in oral health behaviors such as daily frequency of tooth brushing and oral health status such as BOP, decayed teeth, filled teeth, and the OHI-S score. Previous studies showed that clinical oral status (decayed teeth, filled teeth, and gingival status such as BOP and PPD) was a predictor of SROH [1,6,13,14,15,16,17,33,34,35]. Thus, the good SROH seen in the later years might be associated with perceived improvements in oral health status and oral health behaviors.

During the study period, the students who underwent regular dental check-ups had higher probabilities of reporting good oral health than those without. This result is similar to the previous studies suggesting that regular dental check-ups are associated with SROH [1,36]. However, a significant interaction effect between regular dental check-ups and survey year was observed, which indicated that the regular dental check-up effect varied according to the survey year. The predicted probabilities of reporting good SROH suggested that the difference in SROH between students with and without regular dental check-ups decreased from 2011 to 2019. Compared to 2011, the percentage of students with regular dental check-ups increased in 2019. Thus, as more students had regular dental check-ups, the impact of regular dental check-ups on SROH diminished (Table 5 and Figure 2).

In agreement with some studies [1,6,13,14,15,16,17,33,34], students without decayed and filled teeth had better SROH in the present study. Of the independent variables related to SROH, the number of decayed and filled teeth showed the highest OR (Table 5). This consistent association was confirmed when performing logistic regression analysis for each year. The interaction effects between the number of decayed and filled teeth and survey year were not observed in the final analysis, which indicated the strong and consistent association of decayed and filled teeth with SROH during the survey year. A high DMFT score is correlated with anxiety [37], and anxiety affects SROH [2]. Thus, anxiety associated with caries might strongly influence SROH. In addition, dental caries and related complications cause pain and result in visible tooth defects. Therefore, the increasing trend of caries-free students might be strongly associated with the improving trend in SROH.

The present study found that SROH was associated with the daily frequency of tooth brushing and the OHI-S score. These results were partly similar to those of other studies that found that inadequate tooth brushing was associated with poor SROH [5,18,38]. According to previous studies, tooth brushing, the use of dental floss, and regular dental check-ups were significantly associated with good oral hygiene status in Japanese university students [25]. Therefore, the OHI-S score, which indicates clinical oral hygiene status, might affect SROH.

SROH was associated with clinical gingival status (BOP). Previous research showed that SROH and self-reported periodontal health were significantly associated with BOP and the number of deep periodontal pockets [13,17,35]. Those reports supported the present finding. In addition, previous studies indicated a significant association between oral hygiene status and BOP in university students [25,39]. Gingivitis is caused by the continuous accumulation of dental plaque [40]. Although OHI-S and BOP may have mutual effects on SROH, further studies are needed to investigate how accurately university students recognize oral hygiene and gingival bleeding and how that recognition is reflected in SROH.

In addition, in the present results, the CPI score did not show a significant effect on SROH. This result was partly consistent with previous studies in which periodontal variables such as PPD did not agree well with self-assessment [3,41]. A simple question, “In general, how do you consider your oral health?” was used in the present study. It may be more unreliable than a detailed question, such as “Have you ever been told by a dentist that you have gum-related problems?” [17,42]. The more detailed question may lead to students’ self-assessments of periodontal status conforming more to clinical results.

Contrary to some previous studies [13,17], female students reported that their oral health was good more than male students. This result is not surprising given that female students were significantly more likely to have high daily frequency of tooth brushing, regular dental check-ups, no BOP, and a low OHI-S score (chi-squared test, all *p* < 0.001). The present results were consistent with some previous results [6,29,32,43,44]. Potential mechanisms have been proposed to explain differences of health status between the sexes, such as differences in lifestyle and access to dental care [32]. However, the mechanisms remain unclear. Further research will be needed to investigate the contribution of sex to oral health inequality.

The results of this study might be clinically beneficial. Understanding factors related to SROH helps dentists determine what aspects of oral health are important for people [45]. Because SROH may affect quality of life and oral health-related quality of life, factors related to SROH should be controlled at an early life stage to ensure the health of students [1,3,9,46,47]. Not only early detection of dental diseases, but also control of SROH might contribute to improving the quality of life of young people.

This study had several limitations. First, this was a cross-sectional study. To develop a better model, a prospective cohort study and an interventional study would be needed. Further studies will be important to investigate whether the improvement of factors related to SROH contributes to the improvement of SROH or quality of life. Second, factors that might be associated with SROH, such as socioeconomic status [14,48], social capital [2], psychosocial factors [6,49], and systemic conditions [50,51], were not considered. In addition, according to the Japanese School Health Survey in 2019 [52], there are differences in the proportion of students aged 5–17 years with decayed teeth among the surveyed regions. However, we did not investigate the hometown of the participants. Future studies are needed to examine their effects. Third, only ten teeth were examined in the gingival examination because of time constraints, which might have led to under- or over-estimation. There may be a bias that a full examination of all teeth for BOP and PPD could avoid. Fourth, the existence of subjective oral symptoms, such as pain due to dental caries or temporomandibular joint disorders, was not investigated. However, in the present study, few students actually consulted the dentist about oral pain in the oral examination setting, which suggests that the present results may be relatively unaffected by this limitation. Fifth, we did not conduct a transgender-sensitive questionnaire. A previous study showed that oral health in a transgender group was different [53]. Thus, future studies are needed to examine their associations. Finally, all participants were recruited from Okayama University students. This might limit the ability to extrapolate these findings to the general population.

## 5. Conclusions

The percentage of students with very good or good SROH increased from 2011 to 2019. Very good, good, or fair SROH was significantly associated with survey year, and an interaction effect between survey year and regular dental check-ups (year × regular dental check-ups) was observed in university students. Japanese university students who were female, had high daily frequency of tooth brushing, no BOP, no decayed teeth, no filled teeth, and a low OHI-S score were significantly more likely to report very good, good, or fair SROH. The improvement trend in SROH might be associated with changes in oral health behaviors and oral health status.

## Figures and Tables

**Figure 1 ijerph-19-13580-f001:**
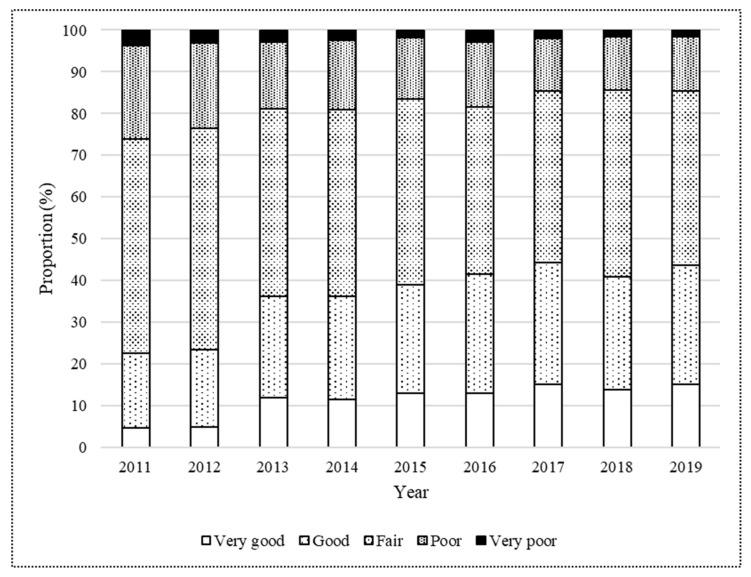
Self-rated oral health status of Japanese university students from 2011 to 2019.

**Figure 2 ijerph-19-13580-f002:**
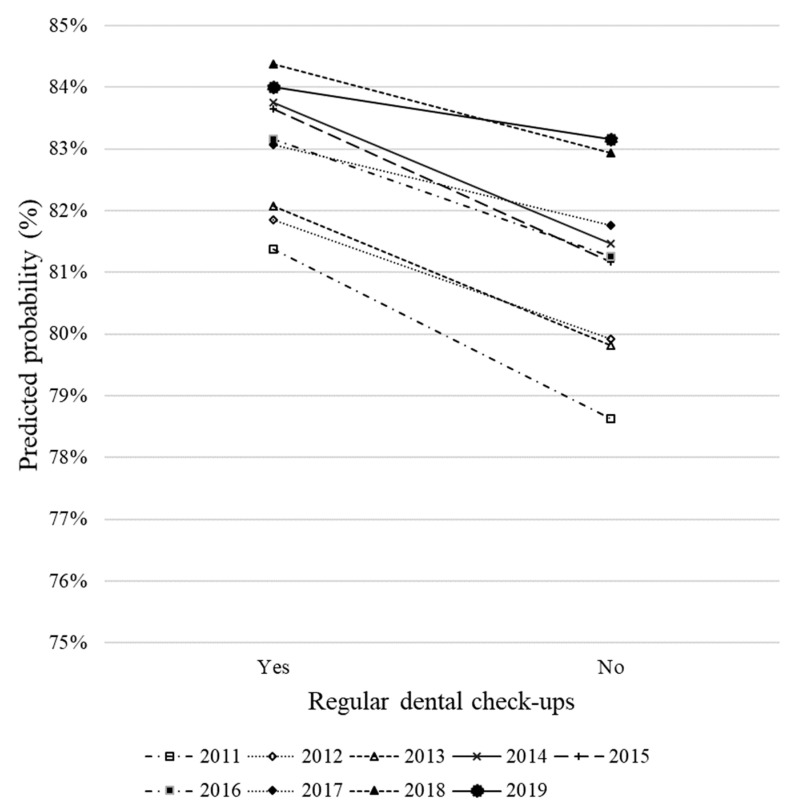
Predicted probability of reporting very good, good, or fair self-rated oral health status in relation to regular dental check-ups and survey year; when sex, tooth brushing (daily frequency), bleeding on probing (BOP), decayed teeth, filled teeth, and Oral Hygiene Index-Simplified (OHI-S) were selected as independent variables in Japanese university students from 2011 to 2019.

**Table 1 ijerph-19-13580-t001:** Participants’ characteristics (*n* = 17,996), Japanese university students, 2011–2019.

Parameter		Year								
		2011	2012	2013	2014	2015	2016	2017	2018	2019
Number of people surveyed		2077 ^1^	1974	2025	2053	2016	1958	1974	1900	2019
Age (y)	18	1752 (84.4) ^2^	1637 (82.9)	1613 (79.7)	1609 (78.4)	1646 (81.6)	1635 (83.5)	1620 (82.1)	1603 (84.4)	1666 (82.5)
	19	325 (15.6)	337 (17.1)	412 (20.3)	444 (21.6)	370 (18.4)	323 (16.5)	354 (17.9)	297 (15.6)	353 (17.5)
Sex	Male	1173 (56.5)	1117 (56.6)	1162 (57.4)	1172 (57.1)	1153 (57.2)	1057 (54.0)	1111 (56.3)	1041 (54.8)	1156 (57.3)
	Female	904 (43.5)	857 (43.4)	863 (42.6)	881 (42.9)	863 (42.8)	901 (46.0)	863 (43.7)	859 (45.2)	863 (42.7)
Tooth brushing (daily frequency)	≤Once	392 (18.9)	388 (19.7)	385 (19.0)	283 (13.8)	234 (11.6)	218 (11.1)	295 (14.9)	176 (9.3)	205 (10.2)
	≥Twice	1685 (81.1)	1586 (80.3)	1640 (81.0)	1770 (86.2)	1782 (88.4)	1740 (88.9)	1679 (85.1)	1724 (90.7)	1814 (89.8)
Use of dental floss	No	1976 (95.1)	1860 (94.2)	1925 (95.1)	1788 (87.1)	1752 (86.9)	1665 (85.0)	1437 (72.8)	1417 (74.6)	1436 (71.1)
	Yes	101 (4.9)	114 (5.8)	100 (4.9)	265 (12.9)	264 (13.1)	293 (15.0)	537 (27.2)	483 (25.4)	583 (28.9)
Regular dental check-ups	No	1777 (85.6)	1562 (79.1)	1741 (86.0)	1706 (83.1)	1478 (73.3)	1408 (71.9)	1427 (72.3)	1343 (70.7)	1346 (66.7)
	Yes	300 (14.4)	412 (20.9)	284 (14.0)	347 (16.9)	538 (26.7)	550 (28.1)	547 (27.7)	557 (29.3)	673 (33.3)
Self-rated oral health (SROH)	Very good	98 (4.7)	99 (5.0)	241 (11.9)	236 (11.5)	261 (12.9)	257 (13.1)	301 (15.2)	262 (13.8)	305 (15.1)
	Good	370 (17.8)	364 (18.4)	495 (24.4)	508 (24.7)	526 (26.1)	558 (28.5)	575 (29.1)	515 (27.1)	575 (28.5)
	Fair	1067 (51.4)	1046 (53.0)	910 (44.9)	920 (44.8)	898 (44.5)	782 (39.9)	810 (41.0)	851 (44.8)	841 (41.7)
	Poor	466 (22.4)	404 (20.5)	324 (16.0)	338 (16.5)	294 (14.6)	308 (15.7)	248 (12.6)	242 (12.7)	265 (13.1)
	Very poor	76 (3.7)	61 (3.1)	55 (2.7)	51 (2.5)	37 (1.8)	53 (2.7)	40 (2.0)	30 (1.6)	33 (1.6)
Bleeding on probing (BOP)	No	485 (23.4)	349 (17.7)	392 (19.4)	389 (18.9)	366 (18.2)	224 (11.4)	454 (23.0)	530 (27.9)	537 (26.6)
	Yes	1592 (76.6)	1625 (82.3)	1633 (80.6)	1664 (81.1)	1650 (81.8)	1734 (88.6)	1520 (77.0)	1370 (72.1)	1482 (73.4)
Probing pocket depth (PPD)	≤3 mm	1850 (89.1)	1806 (91.5)	1766 (87.2)	1737 (84.6)	1578 (78.3)	1511 (77.2)	1469 (74.4)	1490 (78.4)	1483 (73.5)
	≥4 mm	227 (10.9)	168 (8.5)	259 (12.8)	316 (15.4)	438 (21.7)	447 (22.8)	505 (25.6)	410 (21.6)	536 (26.5)
Decayed teeth score	0	1739 (83.7)	1801 (91.2)	1781 (88.0)	1882 (91.7)	1810 (89.8)	1812 (92.5)	1776 (90.0)	1747 (91.9)	1854 (91.8)
	≥1	338 (16.3)	173 (8.8)	244 (12.0)	171 (8.3)	206 (10.2)	146 (7.5)	198 (10.0)	153 (8.1)	165 (8.2)
Filled teeth score	0	917 (44.2)	930 (47.1)	911 (45.0)	959 (46.7)	987 (49.0)	951 (48.6)	1006 (51.0)	977 (51.4)	1141 (56.5)
	≥1	1160 (55.8)	1044 (52.9)	1114 (55.0)	1094 (53.3)	1029 (51.0)	1007 (51.4)	968 (49.0)	923 (48.6)	878 (43.5)
Oral Hygiene Index-Simplified (OHI-S)		0.67 ^3^	0.67	0.50	0.33	0.33	0.50	0.33	0.33	0.33
		(0.17, 1.00)	(0.33, 1.17)	(0.17, 1.00)	(0.00, 0.67)	(0.17, 0.67)	(0.17, 0.83)	(0.00, 0.83)	(0.00, 0.67)	(0.00, 0.67)

^1^ Number of people; ^2^ Number of people (Percentage); ^3^ Median (25th Percentile, 75th Percentile).

**Table 2 ijerph-19-13580-t002:** Differences in age, sex, oral health behaviors, and oral status by self-rated oral health group, Japanese university students, 2011–2019.

Parameter		Self-Rated Oral Health		*p*-Value ^1^
		Poor	Good	
		(*n* = 3325)	(*n* = 14,671)	
Age (y)	18	2746 (82.6) ^2^	12,035 (82.0)	0.452
Sex	Male	1992 (59.9)	8150 (55.6)	<0.001
Tooth brushing (daily frequency)	≥Twice	2610 (78.5)	12,810 (87.3)	<0.001
Use of dental floss	Yes	453 (13.6)	2287 (15.6)	0.004
Regular dental check-ups	Yes	623 (18.7)	3585 (24.4)	<0.001
Bleeding on probing (BOP)	No	553 (16.6)	3173 (21.6)	<0.001
Probing pocket depth (PPD)	≤3 mm	2686 (80.8)	12,004 (81.8)	0.162
Decayed teeth score	0	2639 (79.4)	13,563 (92.4)	<0.001
Filled teeth score	0	1100 (33.1)	7679 (52.3)	<0.001
Oral Hygiene Index-Simplified (OHI-S)	≤0.33	1378 (41.4)	7774 (53.0)	<0.001

^1^ Chi-squared test; ^2^ Number of people (percentage).

**Table 3 ijerph-19-13580-t003:** Logistic regression analysis when independent variables were selected based on the *p*-value (<0.20) by the chi-squared test, Japanese university students, 2011–2019.

Parameter		OR	95%CI	*p*-Value
Sex	Female	1		
	Male	0.878	0.810–0.951	0.001
Tooth brushing (daily frequency)	≤Once	1		
	≥Twice	1.73	1.57–1.92	<0.001
Regular dental check-ups	No	1		
	Yes	1.22	1.11–1.35	<0.001
Bleeding on probing (BOP)	Yes	1		
	No	1.16	1.04–1.29	0.008
Decayed teeth score	≥1	1		
	0	2.59	2.33–2.88	<0.001
Filled teeth score	≥1	1		
	0	2.08	1.92–2.26	<0.001
Oral Hygiene Index-Simplified (OHI-S)	>0.33	1		
	≤0.33	1.43	1.31–1.55	<0.001

CI, confidence interval; OR, odds ratio. Independent variables: sex, tooth brushing (daily frequency), use of dental floss, regular dental check-ups, decayed teeth score, filled teeth score, OHI-S.

**Table 4 ijerph-19-13580-t004:** Factors associated with self-rated oral health (very good, good, and fair) on logistic regression analyses on a yearly basis from 2011 to 2019, Japanese university students.

Parameter		Year								
		2011	2012	2013	2014	2015	2016	2017	2018	2019
Sex	Female			1	1					
	Male			0.769 0.034	0.684 0.002					
Tooth brushing (daily frequency)	≤Once	1	1	1	1	1	1	1	1	1
	≥Twice	1.60 ^1^ <0.001 ^2^	2.10 <0.001	1.63 <0.001	1.73 <0.001	2.41 <0.001	2.06 <0.001	0.625 0.024	1.72 0.007	1.84 0.001
Regular dental check-ups	No	1							1	
	Yes	1.44 0.023							1.57 0.006	
Bleeding on probing (BOP)	Yes			1						
	No			1.44 0.041						
Decayed teeth score	≥1	1	1	1	1	1	1	1	1	1
	0	2.15 <0.001	2.34 <0.001	2.58 <0.001	2.94 <0.001	2.90 <0.001	2.68 <0.001	2.72 <0.001	3.18 <0.001	2.55 <0.001
Filled teeth score	≥1	1	1	1	1	1	1	1	1	1
	0	1.88 <0.001	2.16 <0.001	1.56 <0.001	2.21 <0.001	1.90 <0.001	2.50 <0.001	2.11 <0.001	2.18 <0.001	2.19 <0.001
Oral Hygiene Index-Simplified (OHI-S)	>0.33	1	1	1	1	1	1	1	1	1
	≤0.33	1.44 0.001	1.42 0.003	1.33 0.025	1.36 0.011	1.33 0.023	1.37 0.011	1.54 0.001	1.50 0.003	1.31 0.036

^1^ Odds ratio; ^2^ *p*-value. Independent variables: sex, tooth brushing (daily frequency), regular dental check-ups, decayed teeth score, filled teeth score, OHI-S. Independent variables not selected by the forward selection method are shown blank.

**Table 5 ijerph-19-13580-t005:** Factors associated with self-rated oral health (very good, good, and fair) on logistic regression analyses in 2011–2019, adding survey year and interactions with survey year, Japanese university students.

Parameter		OR	95%CI	*p*-Value
Sex	Female	1		
	Male	0.873	0.805–0.946	0.001
Tooth brushing (daily frequency)	≤Once	1		
	≥Twice	1.68	1.52–1.86	<0.001
Bleeding on probing (BOP)	Yes	1		
	No	1.16	1.04–1.29	0.008
Decayed teeth score	≥1	1		
	0	2.57	2.31–2.86	<0.001
Filled teeth score	≥1	1		
	0	2.05	1.89–2.22	<0.001
Oral Hygiene Index-Simplified (OHI-S)	>0.33	1		
	≤0.33	1.38	1.27–1.50	<0.001
Year		1.06	1.04–1.07	<0.001
Year × Regular dental check-ups		1.03	1.01–1.05	0.002

CI, confidence interval; OR, odds ratio. Independent variables: sex, tooth brushing (daily frequency), regular dental check-ups, decayed teeth score, filled teeth score, OHI-S, survey year, interactions with survey year (year × sex, year × tooth brushing, year × regular dental check-ups, year × BOP, year × decayed teeth score, year × filled teeth score and year × OHI-S).

## Data Availability

All the relevant data are included in the manuscript.

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
