# Peer review of "Trends in Self-Rated Oral Health and Its Associations with Oral Health Status and Oral Health Behaviors in Japanese University Students: A Cross-Sectional Study from 2011 to 2019"

_ijerph, 2022, doi:10.3390/ijerph192013580_

Round 1

Reviewer 1 Report

Dear authors,

congratulations for the interesting topic chosen. To clarify some aspects, please specify the following:

- does the health insurance system in your country require 1-2 checks per year?

- is health education carried out in schools, universities?

-can there be differences relative to the dental status depending on the environment of origin of the students (rural/urban)?

- was the limitation of the examination to certain dental units determined by time factors?

Author Response

Comments and Suggestions for Authors:

Dear authors,

congratulations for the interesting topic chosen. To clarify some aspects, please specify the following:

Our response: Thank you for your comments. We have revised our manuscripts based on your comments as below.

- does the health insurance system in your country require 1-2 checks per year?

Our response: Thank you for your comment. In Japan, the health insurance system does not require dental checks.

- is health education carried out in schools, universities?

Our response: Thank you for your comment. In Japan, dental health education is provided at schools before entering university. University students attend health education lectures if they want.

-can there be differences relative to the dental status depending on the environment of origin of the students (rural/urban)?

Our response: Thank you for your comment. According to the Japanese School Health Survey in 2019, there are differences in the proportion of students aged 5-17 years with decayed teeth among the surveyed regions. However, we did not investigate the hometown of the participants. We have added the comments in Discussion following the reviewer’s suggestion (L346-350).

- was the limitation of the examination to certain dental units determined by time factors?

Our response: Thank you for your comment. As the reviewer pointed out, dental examination was restricted due to limited time. We have added the comments in Discussion following the reviewer’s suggestion (L351).

Reviewer 2 Report

The study aimed  to investigate the trends in self-rated oral health and how oral health behaviors and oral health status are associated with changes in self-rated oral health of Japanese university students.

The study is well conducted and the manuscript reads well. I have only a few comments to improve the clarity of the manuscript:

1- have the authors considered assessing the medical status of the enrolled participants? This might be a limitation to the current study. please make a comment in the discussion.  

2- The inclusion criteria include only subjects aged aged ≥ 20. It is unclear why the authors choose the age of 20 not 18. Please clearly explain. 

3- Did the authors found any transgender participants in their cohort. It is known that oral health in transgender group is different. Please cite the following paper (Manpreet K, Ajmal MB, Raheel SA, Saleem MC, Mubeen K, Gaballah K, Faden A, Kujan O. Oral health status among transgender young adults: a cross-sectional study. BMC Oral Health. 2021 Nov 12;21(1):575. doi: 10.1186/s12903-021-01945-x. PMID: 34772385; PMCID: PMC8588739) and discuss the relevance of that in the discussion section.

Author Response

Comments and Suggestions for Authors:

The study aimed to investigate the trends in self-rated oral health and how oral health behaviors and oral health status are associated with changes in self-rated oral health of Japanese university students.

The study is well conducted and the manuscript reads well. I have only a few comments to improve the clarity of the manuscript:

Our response: Thank you for your comments. We have revised our manuscripts based on your comments as below.

1- have the authors considered assessing the medical status of the enrolled participants? This might be a limitation to the current study. please make a comment in the discussion. 

Our response: Thank you for your comments. As the reviewer pointed out, we did not assess the medical status of the enrolled participants. This is a limitation of our study. We have added the comments in Discussion following the reviewer’s suggestion (L346).

2- The inclusion criteria include only subjects aged ≥ 20. It is unclear why the authors choose the age of 20 not 18. Please clearly explain.

Our response: Thank you for your comments. Inclusion criteria include students who were 18 or 19 years to avoid age-related factors such as smoking and systemic diseases. We have revised the sentences to avoid misleading following the reviewer’s suggestion (L87, 88).

3- Did the authors found any transgender participants in their cohort. It is known that oral health in transgender group is different. Please cite the following paper (Manpreet K, Ajmal MB, Raheel SA, Saleem MC, Mubeen K, Gaballah K, Faden A, Kujan O. Oral health status among transgender young adults: a cross-sectional study. BMC Oral Health. 2021 Nov 12;21(1):575. doi: 10.1186/s12903-021-01945-x. PMID: 34772385; PMCID: PMC8588739) and discuss the relevance of that in the discussion section.

Our response: Thank you for your comments. We did not conduct a transgender-sensitive questionnaire. We have added the comments in Discussion following the reviewer’s suggestion (L356-359).